# Injuries and Illnesses in Male and Female Sailors Throughout the Professional Sailing Circuit SailGP: A Retrospective Cohort Study of SailGP’s Season 3 [note 1]

**DOI:** 10.3390/jfmk10040394

**Published:** 2025-10-09

**Authors:** Matthew Linvill, Thomas Fallon, Hannah Diamond, Jo Larkin, Neil Heron

**Affiliations:** 1School of Medicine, Royal College of Surgeons in Ireland, Dublin D02 YN77, Ireland; mtlinvill@gmail.com; 2Medical Department, SailGP, London SW7 4ES, UK; 3Fortitude Medical Corporation, San Francisco, CA 93446-2220, USA; 4Centre for Public Health, Queen’s University Belfast BT7 1NN, UK; 5Edinburgh Sports Medicine Research Centre, Institute for Sport, PE and Health Sciences, University of Edinburgh, Edinburgh EH8 9YL, UK; 6Emirates GBR SailGP, Portsmouth PO5 3PA, UK; 7Fortius Clinic, London W1U 2SB, UK

**Keywords:** SailGP, injury, sailing, sport, epidemiology, sex, concussion, mixed-sex

## Abstract

**Objectives:** SailGP is an international professional mixed-sex sailing competition, which uses F50 foiling catamarans capable of reaching speeds up to ~100 km/h. This seminal study assesses injuries and illnesses observed by male and female sailors during trainings and competitions in SailGP’s third season. This study aims to assess injury and illness incidence, comparing results with other professional sailing events and high-performance sports. In addition, injury and illness risk factors (sex and position) will be explored with the goal to reduce morbidity for future seasons. **Materials and Methods**: This retrospective cohort design analysed medical records of male and female sailors during SailGP’s third season (April 2022 to May 2023). Risk factors assessed included sailor sex, sailor position (helm, strategist, grinder, flight controller and wing trimmer), sailing venue, wind speed and mechanism of injury/nature of illness. International Olympic Committee reporting guidelines on injuries and illnesses were followed, including the STROBE-SIIS checklist. Confidence intervals were set at 95%, statistical tests were two-sided and *p*-values < 0.05 were considered statistically significant. **Results:** A total of 40 on-water injuries were reported in 32 athletes. Injury incidence was greater during competitions than trainings, with strategists and then grinders being the most frequently injured positions. Competition injury incidence was 32.6 per 1000 h and 6.42 injuries per 365 days. Training injury incidence was 2.62 injuries per 1000 h and 3.82 injuries per 365 days. Knee, ankle, hand and head injuries were most prevalent, with three concussions observed during trainings and competitions (two female and one male). Direct impacts and falls during manoeuvres caused most injuries. Overall injury incidence (IRR = 2.69 [95% CI 1.41–5.16]), risk of training injuries (RR = 3.75 [95% CI 1.59–8.83], *p* = 0.001), risk of competition injuries (RR = 1.79 [95% CI 0.65–4.90], *p* = 0.25) and overall concussion risk (RR = 10.04 [95% CI 0.91–110.46], *p* = 0.02) were greater in females. Ten sailors accounted for 17 illnesses. Females had a 3.33 increase in training and competition illnesses (IRR = 3.33 [95% CI 0.94–11.81]). **Conclusions:** Competition injury incidence was higher than previous reported sailing studies. Knee injuries were most prevalent and direct impacts caused most injuries. Female sailors reported a higher injury and illness incidence. These results may guide injury prevention efforts and the development of an IOC-equivalent consensus statement. Future studies should examine time loss.

## 1. Introduction

While SailGP is one of the most prominent sailing competitions in the world, little is known about its injury and illness patterns. Often described as the “Formula 1 of sailing,” Sail Grand Prix (SailGP) is a professional, mixed-sex, international sailing competition that uses high-performance F50 foiling catamarans capable of reaching speeds up to approximately 100 km/h [1]. The sailors who compete in SailGP represent the highest level of sailing, with many being Olympic medallists and America’s Cup veterans. While sailing is a relatively safe sport, with energy demands comparable to football [2], these high-performance boats carry significant risks. F50s generate extreme forces during foiling, manoeuvres and transitions, which can create hazards such as collisions, structural failures, capsizes and entrapment. In 2013, these dangers were highlighted when Olympic gold medallist Andrew “Bart” Simpson passed away after a boat similar to the F50, the Artemis AC72, capsized in San Francisco [3].

These risks are amplified by a large crew size, with teams typically consisting of six sailors, five males and one female. The six sailors on board have distinct roles: two grinders (forward and aft) power the boat’s hydraulics, the flight controller manages the hydrofoils’ angle of attack to control ride height, the wing trimmer adjusts the boat’s wing to optimize speed, the driver steers via the rudders and the strategist provides tactical input during the race [4]. In addition, each position has their own unique movement patterns. Strategists often transition before a manoeuvre, ensuring a sailor is available to steer the vessel after a manoeuvre. Wing trimmers and grinders follow quickly after to re-establish F50 power and trim. Finally, after the manoeuvre is complete, the driver transitions when the boat is moving in a straight line.

Transitions involve crossing from one hull to the next across a trampoline surface. This movement requires agility and coordination, as sailors must avoid the F50’s wing and lines while covering ~8 m when the F50 is in motion [5]. The speed and timing of these transitions are critical to performance, as delays or mistakes can result in loss of boat control and/or speed.

Given the significance of SailGP within the sailing community and the physical demands placed on its sailors, it is critical to improve SailGP’s understanding of its the injury and illness patterns. Therefore, the primary objective of this paper is to assess injury and illness patterns in SailGP. This will be achieved using the first three steps of van Mechelen et al.’s “sequence of prevention” [6]. This framework consists of four steps: identify and describe the extent of the sports-related problem (injuries and illnesses); identify the factors and mechanisms that contribute to these problems; introduce measures to reduce injury and illness risk/severity; and evaluate the measures’ effects by repeating the first step. Within this analysis, we sought to identify and understand associations between injuries and illnesses and risk factors, such as sailor sex, position (helm, strategist, grinder, flight controller, wing trimmer), wind speed, limited training time and mechanism of injury/nature of illness [2]. When comparing injury incidence to other professional sailing competitions, we hypothesized that SailGP’s injury incidence would be higher due to the physiological demands of sailing an F50. Furthermore, we hypothesized that female sailors would be at greater risk of injury compared to male sailors due to less training time on the F50 and anatomical/physiological differences such as reduced neuromuscular-control, muscle imbalances between agonists and antagonists and fluctuations in hormone levels [7].

## 2. Materials and Methods

This retrospective cohort study analyses injuries and illnesses sustained by sailors during SailGP’s third season (April 2022 to May 2023). During its third season, 54 active sailors (males n = 45, 83% and females n = 9, 17%) competed in nine teams across a series of 11 Grands Prix with wind speeds ranging from approximately 6–40 km/h [1,8]. Grand Prix locations included: Sydney, Australia; St. Tropez, France; Bermuda; Chicago, USA; Christchurch, New Zealand; Singapore; San Francisco, USA; Dubai, UAE; Cádiz, Spain; Plymouth, UK; and Copenhagen, Denmark. Each Grand Prix was scheduled to have 6 races. Each race was about 12 min long, each F50 had a crew of approximately 6 sailors (usually 5 males and 1 female) and all teams trained for roughly the same amount of time. Injuries and illnesses from training sessions and competitions were stored in the electronic medical record system (EMRS), The Sports Office, Kitman Labs, Dublin, Ireland [9], which is known for providing data management services to some of the world’s largest football leagues. Data uploaded to The Sports Office was used for the analysis in this paper and is available upon reasonable request to SailGP. Documented data included the sailor position (e.g., grinder, strategist, wing trimmer, flight controller or driver) (Figure 1), the occasion of the injury (competition, training or other), the sailing venue, the event causing the injury (during manoeuvre, non-sports participation or other), the sex of the sailor (male or female), the location of the injury and the mechanism of injury/nature of illness. Furthermore, wind speed in knots was obtained from information supplied by local weather stations to Weather Underground [10]. Current International Olympic Committee (IOC) reporting guidelines [11], including the STROBE-SIIS Statement 1.0 and research done by Tan et al. [2], formed the basis of this analysis. Ethical approval for this research was granted through Queen’s University Belfast, Northern Ireland (Faculty REC Reference Number MHLS 24_129, 28 August 2024).

The F50 is crewed by six athletes: the driver, who is responsible for steering and crew coordination; the wing trimmer, who adjusts the wing to optimize power; the flight controller, who manages foiling stability and ride height; two grinders, who power the hydraulics to adjust sails and foils; and the strategist, who provides tactical input.

### 2.1. Implementation and Data Collection

All medical data from SailGP’s third season (April 2022–May 2023) were reviewed for this analysis. SailGP medical staff were required to report all calls for athletes and staff seeking medical attention. Inclusion criteria for this study included:(i)All male and female sailors participating in SailGP;(ii)On-water injuries sustained during competitions;(iii)On-water injuries sustained during training sessions;(iv)All reported illnesses between April 2022 and May 2023.

After data inclusion, injuries and illnesses were screened for duplicates. Furthermore, data that were unclear or incomplete were reviewed and adjusted accordingly to improve data quality following consensus by M.L. and J.L.

### 2.2. Definition of Injury and Illness

An injury was defined as “any new injury or exacerbation of pre-existing injuries which occurred during training and/or competition” in SailGP’s third season [12]. An illness was defined as any acute or chronic condition requiring medical attention between April 2022 and May 2023. Injuries and illnesses were recorded if an athlete sought medical attention, irrespective of severity [12].

### 2.3. Statistical Analysis

Data from the EMRS were processed with a Macintosh computer using either Google Sheets, Microsoft Excel or R to determine injury and illness count, prevalence, incidence, incidence rate ratios (IRR) and risk ratios (RR) between male and female sailors. Incidence, RR and IRR were presented with 95% confidence intervals. Statistical tests were two-sided and *p*-values <0.05 were considered statistically significant. Data analysis was guided by the CHAMP guidelines (Table A5) [13].

#### 2.3.1. Competition Injury Incidence

To assess competition injury incidence, injury incidence per 1000 h and per 365 days was calculated. Competition exposure time was estimated by dividing each Grand Prix into two days: day 1 consisted of three races with nine teams and day 2 consisted of two races with nine teams, plus a final race with three teams. Each race was assumed to last 12 min and each team was assumed to have six sailors. After determining competition sailing exposure (hours or days), the injury incidence was calculated using the number of on-water competition injuries identified on the EMRS (Equations (1) and (2)).(1)$$Comp\;Inj/1000\;hrs=\left(\frac{no.of\;inj}{no.of\;races\times 0.2\;hr\;sailing\times no.of\;teams\times 6\;sailors}\right)\times 1000\;hr$$(2)$$Comp\;Inj/365\;days=\left(\frac{no.of\;inj}{no.of\;days\;sailing\times no.of\;teams\times 6\;sailors}\right)\times 365\;days$$

#### 2.3.2. Training Injury Incidence

Training injury incidence was also calculated per 1000 h and per 365 days. To determine the exposure time, the number of training sessions was multiplied by the number of teams sailing, the number of sailors on board and the duration of each training session. Each training session was assumed to last for four hours, with each team comprising six sailors on board. After calculating exposure time, on-water training injuries were identified using the EMRS and a corresponding injury incidence was calculated (Equations (3) and (4)). It should be noted that on-water training injury incidence does not capture injuries occurring on land or training outside of SailGP, as many sailors participate in additional sailing circuits outside of SailGP during the race season.(3)$$On-water\;Training\;Inj/1000\;hrs=\left(\frac{no.of\;inj}{no.of\;trainings\times 4\;hr\times no.of\;teams\times 6\;sailors}\right)\times 1000\;hr$$(4)$$On-water\;Training\;Inj/365\;days=\left(\frac{no.of\;inj}{no.of\;days\;sailing\times no.of\;teams\times 6\;sailors}\right)\times 365\;days$$

#### 2.3.3. Relative Risk, Incidence Rate Ratios and Statistical Analysis

Relative risk (RR) was calculated using Equation (5) with a 95% confidence interval (CI). Incidence rate ratios (IRRs) were calculated using Equation (6) with a 95% confidence interval (CI).(5)$$Risk\;Ratio=\frac{\left(\frac{inj\;exposed}{total\;inj}\right)}{\left(\frac{inj\;unexposed}{total\;inj}\right)}$$(6)$$Incidence\;Rate\;Ratio=\frac{\left(Incidence\;Rate\;Exposed\right)}{\left(Incidence\;Rate\;Unexposed\right)}$$

### 2.4. Patient and Public Involvement

After the collection and analysis of results, findings were discussed with author Hannah Diamond (H.D.), a female strategist for SailGP’s team Great Britain (GBR) and a trained physiotherapist. H.D.’s experience as a sailor was instrumental in identifying potential injury trends. She was consulted prior to disseminating research to SailGP stakeholders, publishers and conferences.

### 2.5. Equity Diversity and Inclusion

During SailGP’s origins, F50s were built for male sailors. Because males are generally larger in stature than females, author J.L. identified that female sailors experienced increased difficulty exiting the F50 cockpit. Consequently, a step was installed into the F50 cockpit for sailors that are shorter in stature. These findings were discussed during a women sailor’s forum, and J.L. identified concerns raised by female sailors including boat specifications, equipment suitability and training time.

Because SailGP is a mixed-sex sporting event, it provides a unique platform for assessing injuries in professional sport. While most high-performance sports have separate male and female events, in SailGP each team consists of both male and female sailors. This structure allows direct comparison of injury rates and mechanisms between male and female athletes. By including GBR’s strategist (Hannah Diamond) as a co-author and consultant for this research, we were able to explore potential mechanisms contributing to female sailors’ and strategists’ increased injury incidence.

## 3. Results

### 3.1. Injury Demographics

During SailGP’s third season, 54 active sailors competed in a series of 11 Grands Prix (males *n* = 45, 83% and females *n* = 9, 17%). During training sessions and competitions, 40 sailor injuries were reported in 32 different athletes, with 19 occurring during competition and 21 during training. Of the 40 injuries observed, 35% occurred in female sailors (Table A1). Furthermore, strategists and grinders sustained the highest number of injuries (Table A1).

When assessing on-water training and competition injuries per venue, the following percentage of injuries were observed: Sydney, Australia (27.5%); St. Tropez, France (15%); Bermuda (15%); Chicago, USA (10%); Christchurch, New Zealand (7.5%); Singapore (5%); San Francisco, USA (5%); Dubai, UAE (5%); Cádiz, Spain (5%); Plymouth, UK (2.5%); Copenhagen, Denmark (2.5%) (Figure 2). Wind speed showed a significant positive relationship with the number of competition injuries (R^2^ = 0.70, *p* = 0.001). The competition with the highest number of injuries, Sydney, Australia (*n* = 6), also had the highest average maximum windspeed of 40.84 knots (Figure 3).

Injury prevalence by race venue: Sydney, AUS (27.5%, *n* = 11); St. Tropez, FRA (15%, *n* = 6); Bermuda (15%, *n* = 6); Chicago, USA (10%, *n* = 4); Christchurch, NZ (7.5%, *n* = 3); Singapore (5%, *n* = 2); San Francisco, USA (5%, *n* = 2); Dubai, UAE (5%, *n* = 2); Cádiz, ESP (5%, *n* = 2); Plymouth, UK (2.5%, *n* = 1); Copenhagen, DEN (2.5%, *n* = 1)

A significant positive relationship was observed between average maximum wind speed and the number of competition injuries (R^2^ = 0.70, *p* = 0.001). Wind speed data were obtained from local weather stations via Weather Underground [10].

### 3.2. Injury Incidence

Sailors trained for a total of 2004 days and 8016 h. Total training injury incidence was 2.62 injuries per 1000 h and 3.82 injuries per 365 days (Table A2). Sailors competed for a total of 1080 total sailor days and 583.2 total sailor hours. Total competition injury incidence was 32.6 per 1000 h and 6.42 injuries per 365 days (Table A2).

When compared to men, females had a higher overall injury incidence rate ratio (IRR female vs. male = 2.69, 95% CI 1.41–5.16). Likewise, female athletes had a higher relative risk (RR) of training injuries (RR = 3.75, 95% CI 1.59–8.83, *p* = 0.001), competition injuries (RR = 1.79, 95% CI 0.65–4.90, *p* = 0.25) and overall concussion risk (RR = 10.04, 95% CI 0.91–110.46, *p* = 0.02), although the RR for competition injuries was not statistically significant (*p* > 0.05).

Overall female injury incidence per 365 days was 8.87 (95% CI 2.15–4.83) and overall male injury incidence per 365 days was 3.30 (95% CI 2.15–4.83) (Table A2). When comparing male and female training injury incidence per 1000 h, female athletes’ incidence was nearly three times higher than that of male athletes (Table A2). Furthermore, female competition injury incidence was nearly double that of male sailors (Table A2).

Regarding sailor position, strategists had the highest overall injury incidence per 365 days (1.66), followed by grinders (1.54), flight controllers (0.59), wing trimmers (0.71) and drivers (0.24).

### 3.3. Mechanism of Injury

Mechanism of injury for trainings, competitions, male sailors and female sailors are summarized in Table 1. Furthermore, mechanisms by sailor position are detailed in Table 1. Falls during manoeuvres and direct impacts were the most common mechanisms in both males and females during training and competition. The most common mechanisms for strategists and grinders, the most frequently injured positions, were also falls during manoeuvres and direct impacts. One unusual mechanism, not listed in Table 1, was a lightning strike that occurred after a race while returning to port.

### 3.4. Injury Location

During training and competitions, the knee was the most frequently injured location (*n* = 9, 23%), (Table A3). The most common injury locations for male sailors were knee (*n* = 6, 23%), head (*n* = 3, 11.5%), shoulder (*n* = 3, 11.5%) and hand (*n* = 3, 11.5%). In contrast, the most common injury locations for female sailors were the ankle (*n* = 3, 21%) and the knee (*n* = 3, 21%), (Figure 4). During the course of trainings and competitions, three concussions were observed (two female and one male).

Coloured regions indicate the proportion of reported injuries at each anatomical site: red (>20%), orange (10–20%), yellow (5–10%), green (0–5%) and blue (0% or not applicable). Differences in injury distribution between male and female athletes include a higher prevalence of head, neck, shoulder and hand injuries among male athletes and ankle injuries among female athletes.

When assessing injuries in grinders and strategists, the two most frequently injured positions, strategists most commonly sustained injuries to the knee (*n* = 3, 21%) and ankle (*n* = 3, 21%), while grinders most commonly sustained injuries to the knee (*n* = 4, 31%) and chest (*n* = 2, 15.4%) (Table A3).

### 3.5. Medical Illnesses

Ten sailors accounted for a total of 17 illnesses (4 females [24%] and 13 males [76%]). Two illnesses occurred during competition, eight during training and seven outside of training and competition. Illnesses by location were as follows: Bermuda (six); Dubai, UAE (three); Chicago, USA (two); Copenhagen, DEN (two); Plymouth, UK (two); Singapore (one); Christchurch, NZ (one). The most common categories of illness symptom clusters were upper respiratory (*n* = 4, 30.8%), gastrointestinal (*n* = 3, 23.1%) and dermatological (*n* = 3, 23.1%), (Table A4). The overall training illness rate per 365 days was 1.46 (95% CI 0.45–2.47) and the overall competition illness rate per 365 days was 0.68 (95% CI 0–1.61). The combined competition and training illness rate per 365 days was 1.18. Compared to males, females had a higher combined training and competition illness incidence (IRR female vs. male = 3.33 (95% CI 0.94–11.81)).

## 4. Discussion

### 4.1. Clinical Implications

#### 4.1.1. Injury Incidence

This study focused on the first three steps of van Mechelen et al.’s sequence of prevention and retrospectively analysed injury data from SailGP’s third season (April 2022 to May 2023). When comparing SailGP’s overall injury incidence per 1000 h to other sailing venues, incidence was similar at 4.65 (95% CI 3.21–6.09) (Table A2). Indeed, Nathanson et al., reported that injury rates per 1000 h were 9 for an amateur around-the-world race, 2.2 for the 2003 America’s Cup and 0.59 for an international 2014 Olympic-class regatta [14].

When comparing SailGP’s competition and training injury incidence per 1000 h to other sailing venues, competition incidence was markedly higher at 32.6 (95% CI 17.9–47.2), whereas training incidence was similar at 2.62 (95% CI 1.5–3.74) (Table A2). SailGP’s increased competition injury incidence is likely due to improvements in hydrofoil technology, which allow F50’s to operate at higher speeds [15]. Additionally, many SailGP sailors compete in multiple sailing circuits outside SailGP throughout the season, resulting in higher cumulative load compared to sailors specializing in a single format of competition.

Compared to a selection of year-round, high-performance sports, SailGP’s competition injury incidence remains high, second only to UK Premiership rugby (Figure 5). Indeed, SailGP has a larger competition/match injury incidence (per 1000 h) than the Union of European Football Associations/UEFA, 32.6 (95% CI 17.9–47.2) compared to 23.8. In contrast, SailGP’s training injury incidence is slightly lower (2.62, 95% CI 1.5–3.74) compared to 3.4 [16]. SailGP’s higher competition injury incidence compared to other professional sports may also reflect limited pre-competition training. While some sailors attended a multi-week pre-season training, many had only one to two days to familiarize themselves with the F50, resulting in a much lower training to competition time ratio than most other professional sports. While SailGP and UEFA have higher injury rates during competitions, some sports have higher injury rates during trainings. For example, downhill mountain bikers may have increased training injury rates due to the bikers experiencing the track for the first time [17].

Injury incidence per 1000 h across multiple sailing events and a selection of high-performance sports [14,16,18,19].

#### 4.1.2. Illness Incidence

In contrast to injury incidence, illness incidence per 365 days was lower during competitions than during trainings, 0.68 (95% CI 0–1.61) versus 1.46 (95% CI 0.45–2.47). These findings may reflect illness incubation periods or underreporting by sailors; however, additional data are needed to draw definitive conclusions.

#### 4.1.3. Mechanism of Injury

During training and competition, the most common injury mechanisms were direct impacts (*n* = 15, 38%) and falls during manoeuvres (*n* = 11, 28%), (Table 1). These results align with research done by Tan et al., who found that the most common mechanisms of injury during the 2014 Sailing World Championships were hitting a part of the boat (42%) and tripping/falling (39%) [2]. Direct impacts are likely due to the unstable nature of an F50 which can lead to rapid and unexpected decelerations when an F50 crashes into the water (Figure A1). Furthermore, falls during manoeuvres are likely caused by frequent changes in boat direction (tacking and gybing), during which sailors must transition across the two hulls of the boat on an unsteady trampoline surface at high speeds (Figure A2). One additional injury mechanism, not reported in Table 1, was a lightning strike that occurred after a race when a boat was returning to port. To prevent similar incidents in the future, SailGP has implemented a new protocol for adverse weather conditions.

#### 4.1.4. Injury Location

When assessing injury location, the knee was the most frequently injured site (*n* = 9, 23%) Table A3. These results contrast with a review of sailing injuries done by Nathanson et al., who examined non-foiling boats and found that upper extremity injuries were most prevalent, followed by lower extremity and chest injuries [14]. SailGP’s increased proportion of lower extremity injuries is likely due to the dynamic nature of sailing a foiling catamaran, which requires multiple transitions across the two hulls. Additional common injuries included head (10%, *n* = 4) and neck (7.5%, *n* = 3) injuries. The increased prevalence of head and neck injuries may be attributed to rapid and unexpected decelerations when an F50 crashes into the water (Figure A1).

Because foiling catamarans present a heightened risk of head injuries, including concussions, SailGP has developed the first bespoke sailing concussion assessment. This protocol was developed internally using the SCAT6 and the sailing medical expertise of the team [20]. The protocol is currently in the process of being submitted for publication in a peer-reviewed journal.

#### 4.1.5. Male vs. Female Sailors

Compared to men, females had a higher overall injury incidence (IRR female vs. male = 2.69, 95% CI 1.41–5.16). Likewise, female athletes had a higher RR of training injuries (RR = 3.75, 95% CI 1.59–8.83, *p* = 0.001), competition injuries (RR = 1.79, 95% CI 0.65–4.90, *p* = 0.25) and overall concussion risk (RR = 10.04, 95% CI 0.91–110.46, *p* = 0.02), although RR for competition injuries was not statistically significant (*p* > 0.05).

These trends mirror findings by Tan et al., who reported that females had roughly twice the injury prevalence of males in the mixed pair class Narca 17, where one male and female sailor must be on board [2]. Females’ increased injury incidence may be due to several factors. First, female sailors may have an increased risk for knee injuries due to anatomical and physiological differences (e.g., reduced neuromuscular-control, imbalance of muscle agonists and antagonists and fluctuations in hormone status) [7]. Females may also have an increased risk for ankle injuries due to changes in hormonal status and neuromuscular control, although further research is needed [21]. These factors may explain why the most common injury locations for female sailors were the knee (*n* = 3, 21%) and ankle (*n* = 3, 21%), compared to males, who most frequently injured the knee (*n* = 6, 23%), head (*n* = 3, 11.5%), shoulder (*n* = 3, 11.5%) and hand (*n* = 3, 11.5%) (Table A3).

Lastly, female sailors’ increased risk of concussions may be due to a reduced head-neck segment mass, which can increase the angular acceleration of the head during impact, or hormonal effects of oestrogen on cerebral blood flow [22]. However, additional research is needed to clarify the relationship between sailor sex and concussion risk, as the sample size in this paper was small.

When assessing upper extremity injuries in males, the higher prevalence of shoulder and hand injuries is likely positional. During SailGP’s third season, most grinders, wing trimmers and drivers were male. These positions place increased demands on the upper limbs through manually powering the F50 with a hand-crank (grinding pedestal), trimming the sails of the F50 and driving the boat.

In addition to physiological factors, equipment design may also contribute to increased injury risk for female sailors. Because SailGP sailing equipment was originally designed for male sailors, female sailors may experience poorly fitting gear, or poorly tailored equipment. For example, at the beginning of season three, several female sailors had greater difficulty exiting the boat’s cockpit due to shorter stature. This hindered their ability to achieve a safe position during potential crashes and rapid changes in boat direction.

Finally, female sailors’ increased injury risk may be related to reduced F50 exposure during SailGP’s third season. At the start of the season teams were required to rotate female sailors on their roster. Because each team only had one active female sailor, their total F50 exposure time was significantly lower than that of male sailors. This resulted in reduced familiarity with SailGP equipment and procedures. Near the end of season three, this rotation rule was removed.

In addition to an increased injury incidence, female sailors also had a higher overall illness incidence (IRR female vs. male = 3.33, 95% CI 0.94–11.81). These findings align with research done by Crunkhorn et al., who reported that females in an Australian Olympic class and State Sailing Pathway Program (SSPP) experienced a 3.6-fold increase in illnesses compared to males (IRR = 3.6, 95% CI 2.0–6.8) [23]. This result warrants further investigation to determine the underlying factors contributing to the increased rate of reported illnesses in female sailors.

#### 4.1.6. Positional Differences

During training and competition, strategists and grinders experienced the highest number of injuries (Table A1). Strategists had the highest overall injury incidence per 365 days (1.66), followed by grinders (1.54), flight controllers (0.59), wing trimmers (0.71) and drivers (0.24). These results align with Tan et al., who found that in two-person classes, the helmsmen (i.e., driver of the boat) had a slightly lower injury prevalence compared to the crew, although the difference was not statistically significant (*p* = 0.655) [2].

Strategists’ increased injury incidence may be explained by several factors. Firstly, strategists are typically female and shorter in stature, which may impede exiting the cockpit and reaching safe positions during manoeuvres. Second, strategists transition before manoeuvres and do not have an active role in turning the boat, reducing their ability to predict sudden changes in F50 acceleration. Third, strategists’ safety tethers have more slack because they are not deflected around the wing of the boat (Figure A2). This may result in longer collisions and contrasts with other positions on the F50, except for the driver, who typically crosses the boat at slower speeds, when the F50 is traveling in a straight line. These observations correlate with the fact that the majority of strategist injuries were caused by falls during manoeuvres and direct impacts (Table 1). Strategists most commonly injured the knee (*n* = 3, 21%) and ankle (*n* = 3, 21%) (Table A3). Finally, nearly all female sailors in season three were strategists, contributing to the higher injury proportion in this position.

In contrast, grinders’ increased proportion of injuries may be related to the forward grinder facing away from the F50’s direction of travel, limiting their ability to anticipate crashes or sudden directional changes. Grinders most frequently injured the knee (*n* = 4, 31%) and chest (*n* = 2, 15.4%) (Table A3) and were more likely to experience injuries from falls during manoeuvres and direct impacts (Table 1). Additionally, the physical demands of the grinder, which includes manually powering the hydraulics of the F50 with a pedestal, may further increase injury prevalence. During season three, 23% of grinder injuries involved the upper extremities (Table A3).

#### 4.1.7. Illnesses

The most common illness symptom clusters were upper respiratory (*n* = 4, 30.8%), gastrointestinal (*n* = 3, 23.1%) and dermatological (*n* = 3, 23.1%) (Table A4). These findings align with Crunkhorn et al., who reported that respiratory infections were the most common illnesses during an Australian Olympic class and State Sailing Pathway Program (*n* = 22, 40.7%) [23]. Upper respiratory illnesses were likely due to increased exposure to pathogens during travel, close quarters of event venues, significant time zone changes and competition-related stress. Furthermore, environmental factors such as water quality may increase pathogen exposure risk [2]. Gastrointestinal illnesses were likely influenced by similar factors affecting upper respiratory illness, as well as variable hygiene standards at venues and changes in diet and hydration while traveling. Dermatological issues were likely caused by repeated exposure to saltwater and wind, limited shower facilities, elevated stress levels and environmental pathogen exposure.

#### 4.1.8. Sailing Venue

When assessing training and competition injuries per venue, injuries were most common in Sydney, Australia, where 27.5% of the season’s injuries occurred (*n* = 11), (Figure 2). Analysis of the relationship between average maximum wind speed and the number of competition injuries revealed a significant positive correlation (R^2^ = 0.70, *p* = 0.001) (Figure 3). These findings are consistent with Nathanson et al., who reported that wind speeds were a primary cause of injuries in recreational sailors [14].

### 4.2. Strengths and Limitations

The strength of this study is its comprehensive documentation of injuries and illnesses during SailGP’s third season, providing a detailed overview of SailGP’s health burden. However, several limitations warrant consideration.

First, injury incidence calculations assumed that each race lasted 12 min, each boat had a crew of six sailors (five males and one female) and all teams trained for the same amount of time. While necessary for analysis, these assumptions may introduce inaccuracies. Nevertheless, it is important to recognize that this study represents the first attempt to quantify injury incidence within SailGP. Moving forward, SailGP will develop the ability to track time on water for each boat.

Second, during season three, SailGP did not record data on time loss or injury pathology. Time loss is essential for calculating injury severity, a key component of van Mechelen et al.’s first step in the “sequence of prevention” [6]. Furthermore, injury pathology is important for informing targeted prevention strategies. These limitations will be addressed for future seasons.

Third, underreporting of injuries and illnesses is a potential concern. As professional athletes, sailors may underreport to continue competing [23]. This could lead to an underestimation of SailGP’s true injury and illness burden.

Finally, the small sample size and single-season scope of this study may limit generalisability. Future research should address these limitations and build upon the foundation established here, enhancing understanding of sailing-related injuries in professional contexts, particularly within SailGP.

## 5. Conclusions

This is the first study to report injury and illness data within SailGP, with a retrospective analysis of its third season (April 2022 to May 2023). Season three had nine teams, with approximately six sailors per team, across 11 Grands Prix. Injury incidence during competitions was over ten times higher than that of trainings (32.6 [95% CI 17.9–47.2] compared to 2.62 [95% CI 1.5–3.74] per 1000 h). Furthermore, competition injury incidence exceeded that of other reported sailing events, such as the America’s Cup. The positions that sustained the most injuries were strategists and grinders. The most injured locations were the knee, ankle, hand and head. Most injuries were due to direct impacts and falls during manoeuvres. Female competition injury incidence was nearly double that of male sailors (51.44 [95% CI 9.35–96.53] compared to 28.81 [95% CI 13.72–43.9] per 1000 h). Female sailors also had a higher overall illness incidence (IRR female vs. male = 3.33 [95% CI 0.94–11.81]). When assessing the impact of wind speed, there was a significant positive relationship between wind speed and competition injuries (R^2^ = 0.70, *p* = 0.001). The competition with the most injuries, Sydney, Australia, also had the highest average maximum windspeed.

To reduce injuries and illnesses at SailGP, this study recommends the following prevention measures. First, sailors should engage in strength training and warm-ups that involve the entire body and emphasize muscles that support the ankle, knee and shoulder [e.g., FIFA11+]. Second, to reduce concussion risk, sailors should perform neck muscle strengthening during trainings and neck muscle activation before events. Third, sailors should engage in exercises that promote proper landing technique. Fourth, sailors should complete sufficient on-water training prior to competing. Fifth, all sailor equipment should be reviewed for effectiveness by male and female sailors. Sixth, SailGP should install shower facilities at venues to promote better hygiene practices. Finally, SailGP should continue to review protocols for terminating races based on adverse weather, with wind and lightning forecasts being of particular importance.

We believe these prevention measures, in addition to future research, will lead to a significant reduction in SailGP’s injury/illness burden. Furthermore, research from this study may lead towards future injury prevention efforts in sailing and contribute to the development of an IOC consensus for the sport. Future research should include prospective surveillance of SailGP injuries and illnesses, with a detailed analysis of time loss.

## Figures and Tables

**Figure 1 jfmk-10-00394-f001:**
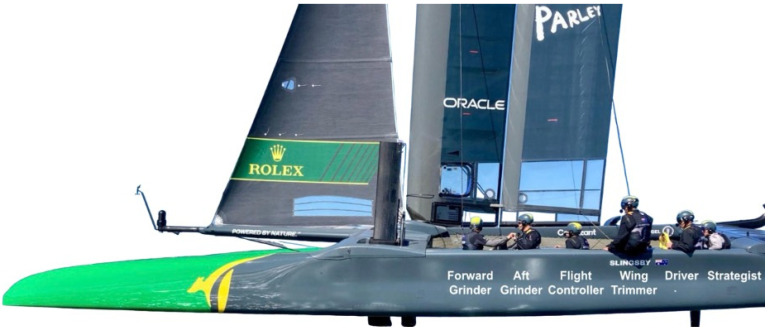
Crew positions on SailGP F50 catamaran.

**Figure 2 jfmk-10-00394-f002:**
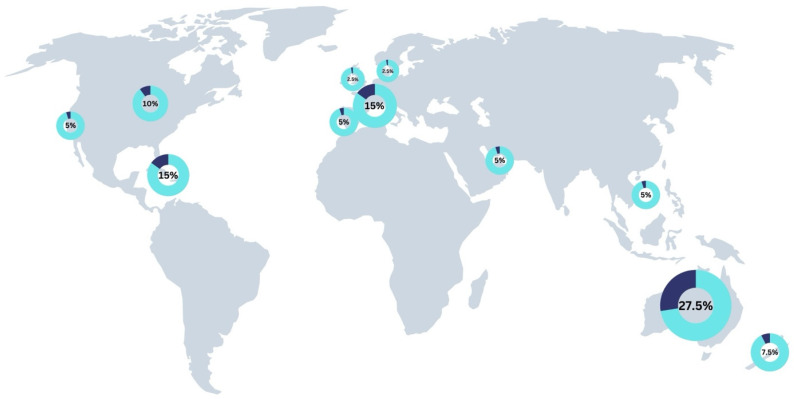
Geographic distribution of on-water training and competition injury prevalence across SailGP Season 3 race venues.

**Figure 3 jfmk-10-00394-f003:**
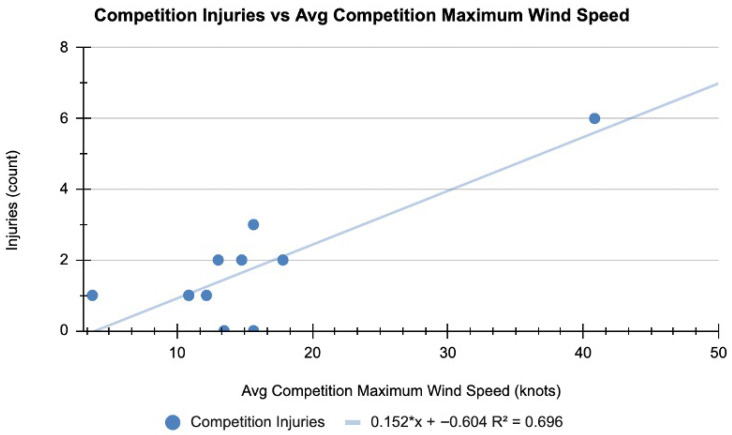
Competition injuries vs. average maximum wind speed.

**Figure 4 jfmk-10-00394-f004:**
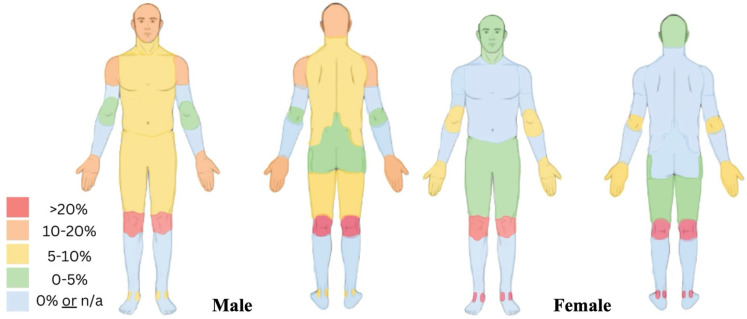
Heatmap of injury location prevalence in male (left) and female (right) SailGP athletes.

**Figure 5 jfmk-10-00394-f005:**
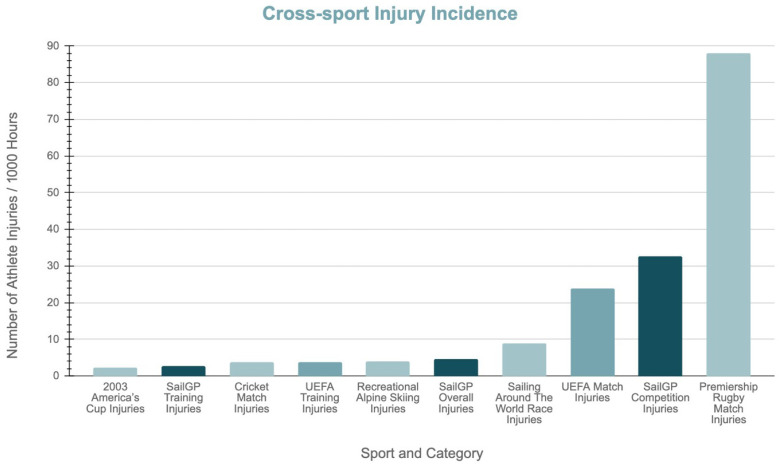
Training and competition cross-sport injury incidence.

**Table 1 jfmk-10-00394-t001:** Percent of injuries by mechanism categorised by training, competition, sailor sex and sailor position. Total injury counts are shown in parentheses. Most strategists were female, whereas grinders, wing trimmers, flight controllers and drivers were predominantly male.

Mechanism of Injury	Total % (*n*)	Training Injuries % (*n*)	Competition Injuries % (*n*)	Male Injuries % (*n*)	Female Injuries % (*n*)	Strategist Injury % (*n*)	Grinder Injury % (*n*)	Wing Trimmer Injury % (*n*)	Flight Controller Injury % (*n*)	Driver Injury % (*n*)
Direct impact	38% (15)	29% (6)	47% (9)	38% (10)	36% (5)	36% (5)	46% (6)	33% (2)	40% (2)	0
Fall during a manoeuvre	28% (11)	24 % (5)	31% (6)	23% (6)	36% (5)	36% (5)	23% (3)	0	40% (2)	50% (1)
Ankle inversion	2.6% (1)	4.8% (1)	0	0	7.1% (1)	7.1% (1)	0	0	0	0
Collision	2.6% (1)	4.8% (1)	0	0	7.1% (1)	7.1% (1)	0	0	0	0
Entering cockpit	5.1% (2)	9.5% (2)	0	3.8% (1)	7.1% (1)	7.1% (1)	0	0	20% (1)	0
Rope-related injury	2.6% (1)	4.8% (1)	0	0	7.1% (1)	7.1% (1)	0	0	0	0
Man overboard	2.6% (1)	0	5.3% (1)	3.8% (1)	0	0	7.7% (1)	0	0	0
Overuse	10% (4)	9.5% (2)	11% (2)	15% (4)	0	0	7.7% (1)	33% (2)	0	50% (1)
Non-contact	2.6% (1)	4.8% (1)	0	3.8% (1)	0	0	7.7% (1)	0	0	0
Unknown	5.1% (2)	9.5% (2)	0	7.7% (2)	0	0	7.7% (1)	17% (1)	0	0

## Data Availability

All study data is freely available upon reasonable request to SailGP. The figures in this manuscript were either produced by the authors of this paper or received from SailGP’s media team. Access and permission for use of SailGP’s photos were formally granted. All copyright procedures have been fully respected.

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
