# Peer review of "Injuries and Illnesses in Male and Female Sailors Throughout the Professional Sailing Circuit SailGP: A Retrospective Cohort Study of SailGP’s Season 3†"

_jfmk, 2025, doi:10.3390/jfmk10040394_

Round 1

Reviewer 1 Report

Comments and Suggestions for Authors

In this manuscript, authors investigated injuries in professional sailing for SailGP, which have been poorly investigated.
The aim was to understand injuries and illnesses observed in sailors during SailGP's third season to improve safety for future seasons.
Authors describe all the scientific information in every chapter of the manuscript with clear, scientific and precise way.
The authors conclude that female sailors had an increase in reported injuries and illnesses; knee injuries were most prevalent, and direct impacts caused most injuries.
These results may guide injury prevention efforts and the development of an International Olympic Committee equivalent consensus statement.
I believe the authors therefore optimally, clearly, precisely, and scientifically achieve their intended results:
• Injury incidence was higher in competitions compared to training.
• Females had a 2.69 increase in reported injuries and a 3.33 increase in reported illnesses.
• Strategists and grinders had the highest proportion of injuries.
• Knee, ankle, hand and head injuries were the most prevalent injuries.
• Falls during maneuvers and direct impacts were the most common mechanisms of injury.
• Upper respiratory, gastrointestinal and dermatological illnesses were the most prevalent.
The manuscript could be published in this form, only after minor English revision

Author Response

✅Reviewer 1

Comment 1:

The English in this paper could be improved to express the research.

Response 1:

Thank you for pointing this out. We agree and have therefore revised the entirety of the paper to improve its ability to express this research. 

Reviewer 2 Report

Comments and Suggestions for Authors

Dear,

Please find my comments attached.

Kind regards

Author Response

Thank you for the comments. Please see the document attached for a response. 

Reviewer 3 Report

Comments and Suggestions for Authors

Dear authors,
All praise for a well-done study and a well-written paper. Sail Grand Prix (SailGP) is a new competition with a relatively new type of boat with great popularity in the world, and such studies are of great importance in the world of sailing. The scientific paper is very thorough and well-written, but I would like to emphasize a few things.
Introduction
The introduction itself is not well-structured and does not present us well enough in the SailGP class. In order to better understand the occurrence of injuries, we need to present the sailing positions (which is done in the last paragraph) with their characteristic movements, i.e. the movements that are the most risky for getting injuries. Type: what are the tasks of the grinder, what are his movements, if he changes the side of the boat when he does so and how much distance does he cover, what is the surface, etc.
Also, this paragraph should not be the last but in the middle of the text. Rephrase the last paragraph and include it in the part of the text where you explain the sailing class I competition.
Line: 63-64, During its third season (April 63 2022 to May 2023), nine teams competed in a series of eleven Grands Prix, with wind 64 speeds ranging from around 6-40 km/h (1,2).
Comment: This sentence should be rephrased or placed in the Materials.
Line 67-68, This study is the first of its kind and aims to 67 understand the injury and illness demands of F50 sailing, with the target of improving 68 safety for future seasons.
There is no need to mention the aim of the study in two places, for clarity of reading it is better when it is all in one place.

Conclusion
I think this section would benefit from a sentence or two about what you recommend should be done to reduce the incidence of injuries among the most vulnerable sailors in SailGP.

Best regards

Author Response

Comment 1 (The introduction is not well-structured and does not present us well enough in the SailGP Class)

1. ✅ Present the sailing positions with their characteristic movements

  • Response: Thank you for this comment, we agree and have therefore added each positions respective movements. This change can be found on page 2 paragraph 3

2. ✅Present relevant tasks, when they change sides of the boat, how much distance they cover, and what is the surface of the boat

  • Response: Thank you for this comment, we agree and have therefore added each positions respective tasks and transition details. This change can be found in paragraph 2 and 3 of the introduction on page 2. 

3. ✅The paragraph that explains SailGP should not be last, but in the middle of the introduction.

  • Response: Thank you for this comment. We agree and have thus relocated the explanation of SailGP earlier in the introduction. This change can be found on page 2.

4. ✅Line: 63-64, should be placed in the materials and methods… “During its third season (April 2022 to May 2023), nine teams competed in a series of eleven Grands Prix, with wind 64 speeds ranging from around 6-40 km/h (1,2)”.

  • Response: Thank you for this response. We agree and have relocated the section. This change can be found on page 3 paragraph 2.

5. ✅There is no need to mention the aim of the study in two places; for clarity of reading, it is better when it is all in one place. Line 67-68, This study is the first of its kind and aims to 67 understand the injury and illness demands of F50 sailing, with the target of improving 68 safety for future seasons.

  • Response: Thank you for the comment. We have removed the aim from the introduction and left the aim in the objectives

✅Comment 2 (I think the conclusion section would benefit from a sentence or two about what you recommend should be done to reduce the incidence of injuries among the most vulnerable sailors in SailGP.)

Response: Thank you for the comment, we completely agree. There is now a paragraph on recommendations for injury and illness prevention in the conclusion. This change can be found on page 17, paragraph 2.

Round 2

Reviewer 2 Report

Comments and Suggestions for Authors

Dear,

Please find my comments attached.

Kind regards

Comments on the Quality of English Language

The quality of the English language is fine.

Author Response

Comment 1 (Ethics Committee Approval):

Please provide documentation confirming the approval of the Ethics Committee for access to and use of data from the Electronic Medical Record (EMR) system.

  • Response: Thank you for this comment. We have uploaded a copy of our ethics document into the MDPI portal. 

Comment 2 (Data Sources):

The manuscript should explicitly state from which websites, databases, or systems the data were obtained. Please clarify whether these data are publicly available, and provide a detailed explanation within the manuscript.

  • Response: Thank you for this comment. We have addressed this comment in paragraph 1 of the methods section on page 3. In addition, there is a statement on data availability on page 18.

Comment 3 (Figures and Copyright):

The manuscript should clearly state whether the figures presented are subject to copyright protection. Authors must specify how access to these figures was obtained and confirm that they hold the necessary permissions for reproduction. Please provide supporting documentation of permissions, where applicable.

  • Response: Thank you for this request. We have added a section to statements on page 18 called Figures and Copyright. In this section, we explain that the figures in this manuscript were either produced by the authors of this paper or received from SailGP’s media team. Access and permission for use of SailGP’s photos were formally granted. All copyright procedures have been fully respected.

Comment 4 (Placement of Figures):

Figure 1 is not appropriately placed in the Introduction. It should be moved either to Section 2 (Materials and Methods) or to the Appendix.

  • Response: Thank you for this comment. We have updated the methods section to include this figure. This change can be found on page 4, after the first paragraph of the methods section.

Comment 5 (Written Statement):

Include in the manuscript a written statement confirming that all ethical standards, privacy protection rules, and copyright procedures have been fully respected.

  • Response: Thank you for this comment. We declared each of these statements in their appropriate sections. These changes can be found on page 18 (Ethical Approval), page 18 (Informed Consent) and page 18 (Figures and copyright). 

Round 3

Reviewer 2 Report

Comments and Suggestions for Authors

Dear,

I am satisfied with the corrections made by the authors. However, I would like to point out several remaining technical issues that need to be addressed in order for the manuscript to be fully ready for publication.

In Section 2 Methods (which should correctly be titled Materials and Methods rather than simply Methods as currently written), revisions are needed for clarity and consistency.

  On page 5, the sentence stating that “the EMR data were processed using a Macintosh computer with Google Sheets, Microsoft Excel, or the R software in order to determine injury and illness counts, prevalence, incidence, incidence rate ratios (IRR), and risk ratios (RR) between male and female players” should be relocated. Since this content overlaps with the description already provided in Subsection 2.3 Statistical Analysis on page 6, it should be integrated there to avoid redundancy. The duplicated text should be removed from Section 2.   In the Conclusion, the inclusion of references is not acceptable. Specifically, the citations to Reference 24 and Reference 6 in parentheses on page 22 should be deleted. The conclusion should focus on summarizing findings and implications rather than citing literature.   Finally, in the Appendix, particularly in Table A2, the legend must clearly define the abbreviations used, such as “Inj/1000 HRS” or “Inj/365 days”, as well as “CI” for Confidence Interval. The same clarification should also be applied to Table A3 to ensure transparency and readability.   Kind regards

Author Response

Comment 1: The Methods section should be renamed as Materials and Methods.

Response 1: Thank you for pointing this out. We agree and have therefore changed the title of the Methods section to Materials and Methods in the abstract and the main body of the paper. These changes can be found on page 1 and page 3.

Comment 2: EMR data processing should be moved/integrated to subsection 2.3 on page 5.

Response 2: Thank you for pointing this out. We agree and have therefore moved and integrated the earlier section regarding data processing to subsection 2.3 on page 5

Comment 3: Do not include references in the conclusion (e.g., page 17). The conclusion should focus on summarizing findings and implications.

Response 3: Thank you for pointing this out. We agree and have therefore removed citations from the conclusion. These changes can be found on page 17.

Comment 4: The appendix (e.g., Table A2 and A3) should define all abbreviations.

Response 4: Thank you for pointing this out. We agree and have therefore revised the entirety of the appendix to make sure all abbreviations are defined. This change can be found on pages 19-21.